# Sub-nanometer mapping of strain-induced band structure variations in planar nanowire core-shell heterostructures

Sara Martí-Sánchez[1,7], Marc Botifoll[1,7], Eitan Oksenberg [2], Christian Koch[1], Carla Borja[1], Maria Chiara Spadaro [1], Valerio Di Giulio [3], Quentin Ramasse [4,5], F. Javier García de Abajo[3,6], Ernesto Joselevich [2] & Jordi Arbiol [1,6✉]

Strain relaxation mechanisms during epitaxial growth of core-shell nanostructures play a key role in determining their morphologies, crystal structure and properties. To unveil those mechanisms, we perform atomic-scale aberration-corrected scanning transmission electron microscopy studies on planar core-shell ZnSe@ZnTe nanowires on α-Al$_2$O$_3$ substrates. The core morphology affects the shell structure involving plane bending and the formation of low-angle polar boundaries. The origin of this phenomenon and its consequences on the electronic band structure are discussed. We further use monochromated valence electron energy-loss spectroscopy to obtain spatially resolved band-gap maps of the heterostructure with sub-nanometer spatial resolution. A decrease in band-gap energy at highly strained core-shell interfacial regions is found, along with a switch from direct to indirect band-gap. These findings represent an advance in the sub-nanometer-scale understanding of the interplay between structure and electronic properties associated with highly mismatched semiconductor heterostructures, especially with those related to the planar growth of heterostructured nanowire networks.

[1] Catalan Institute of Nanoscience and Nanotechnology (ICN2), CSIC and BIST, Campus UAB, Bellaterra, 08193 Barcelona, Catalonia, Spain. [2] Department of Molecular Chemistry and Materials Science, Weizmann Institute of Science, Rehovot 76100, Israel. [3] ICFO-Institut de Ciencies Fotoniques, The Barcelona Institute of Science and Technology, 08860Castelldefels, Barcelona, Spain. [4] SuperSTEM Laboratory, STFC Daresbury Campus, Daresbury WA4 4AD, UK. [5] School of Chemical and Process Engineering & School of Physics and Astronomy, University of Leeds, Leeds LS2 9JT, UK. [6] ICREA, Passeig Lluís Companys 23, 08010 Barcelona, Catalonia, Spain. [7] These authors contributed equally: Sara Martí-Sánchez, Marc Botifoll. ✉email: arbiol@icrea.cat

Over the last three decades, heterostructured semiconductor nanowires (NWs)[1] have been in the spotlight because of their potential as nanoengineered building blocks for a new generation of electronic[2–5], photonic[6] and plasmonic[7,8], devices for quantum[9–12], sensing[13,14], and energy-harvesting[15–19] applications.

The nanometer-scale lateral dimensions and resulting quasi one-dimensional morphology of the NWs facilitate the release of the inherent epitaxial strain, preserving high crystal quality even when interfacing highly mismatched materials[20]. In NWs, plastic and elastic relaxation mechanisms can combine to release strain during axial growth while maintaining a monocrystalline material heterostructure[20,21], thus introducing a broad variety of material combinations in NWs that were not feasible at bulk scales, and further opening up brand new possibilities for electronic band engineering and control over the desired properties[22–28]. In this context, free-standing vertical nanostructures, such as hetero-structured NWs, have been used for three decades as design model systems to study basic physical and quantum phenomena at the nanoscale, offering a high degree of detail and precision[9–11,15,23].

Nonetheless, difficulties arise when trying to interconnect different hybrid NWs to form circuits or networks. Most NW systems to date are grown vertically on planar substrates via the vapor-liquid-solid (VLS) growth mechanism, using a metal liquid catalyst seed[29–33]. Positioning and contacting the grown NWs in circuits/networks is challenging because the NW must often be cut and arranged horizontally, and subsequently contacted to form a circuit. Several attempts have been made to interconnect VLS-grown NWs after or even during the growth process[34,35], but reproducibility and scalability to industry standards remain elusive.

To overcome the scalability issues of VLS while maintaining perfect epitaxy and gaining in reproducibility and robustness, semiconductor NW growth has undergone a dramatic evolution in recent years. The development of new planar growth meth-odologies, such as guided growth (GG)[36,37] and template-assisted selective epitaxy (TASE)[38,39], as well as the more recent appli-cation of selected area growth (SAG)[40,41], allow for the direct growth of horizontal nanowires on top of a selected substrate with high crystal quality together with the possibility of obtaining seamless intersections, consequently enabling the design of complex circuits and networks. These new planar growth stra-tegies are therefore suitable for the large-scale fabrication of tai-lored networks. In particular, guided growth exploits the epitaxial or graphoepitaxial relationship of the material with the substrate in order to guide the VLS growth of nanowires along specific substrate crystal directions or relief features, respectively, hence creating self-assembled arrays of well aligned nanowires. During the last decade, the guided growth of several types of II-VI NWs (such as ZnO[42], CdSe[43], ZnTe[44], or ZnSe[45]) was achieved, eventually leading to the successful assembly of guided core-shell heterostructure arrays involving these materials[46].

This paradigm based on horizontally grown NWs demands a closer look at the strong interactions between the substrate and different materials along the whole length of the grown nanos-tructures. The high strain induced by the lattice mismatch at the interfaces of these horizontal NWs is not as easily released as in vertical ones[20]. The resulting strain profiles can substantially influence the interface properties of these materials and thus also their final emerging electronic, photonic or quantum performance. Therefore, to obtain designated material properties, it is important to monitor in detail the structure of these interfaces and correlate them to the local changes of physical properties.

In particular, crystal lattice deformations caused by strain have been exploited for tuning the electronic band alignment in such material heterostructures[27,28]. These effects are accentuated in radial core-shell nanostructures compared to axial heterostructures

resulting from the increase in the interaction area between the interfaced materials[20,28]. Additionally, along these large interfaces, complex strain relief mechanisms, in which both plastic and elastic deformations occur simultaneously, can take place at the nanoscale[20,21]. The arguments above illustrate that an in-depth examination of the epitaxially induced strain fields at the atomic-scale in these systems is essential to understand and gain control over their (opto)electronic properties.

## Results and discussion

**Materials description: structure, composition, and strain ana-lyses.** In this work, we study the local crystal structure and strain relaxation at the atomic level in horizontal arrays of ZnSe@ZnTe core-shell nanowires grown on $\alpha$-Al$_2$O$_3$ (sapphire) substrates by guided growth, as sketched in Fig. 1a. ZnSe and ZnTe are both direct band-gap semiconductors with room temperature band-gap values of 2.67 eV[47] and 2.26 eV[48,49], respectively, which lie in the visible range[50,51]. These materials form radial type-II p-n heterojunctions when interfaced, which are highly advantageous for photovoltaic applications[52]. The reason for such core-shell combination arises from the fact that ZnTe can be deposited on ZnSe at a lower temperature, making this combination feasible given the growth conditions of both materials[46].

The sapphire substrates were cut in two different orientations (A-plane and C-plane, i.e., $\alpha$-Al$_2$O$_3$ (11$\bar{2}$0) and $\alpha$-Al$_2$O$_3$ (0001), respectively) in order to evaluate different growth directions and plane interactions in this study. n-type ZnSe NWs were grown[45] first and then employed as an epitaxial template (acting as cores) for p-type ZnTe shell growth[44]. The arrays of parallel core-shell NWs follow six equivalent <1$\bar{1}$00> directions in the case of C-sapphire substrates, whereas they follow the two <1$\bar{1}$00> plus four <1$\bar{1}$01> directions in the case of A-sapphire. Details on the growth directions are offered in Supplementary Fig. 1. Our scanning transmission electron microscopy (STEM) analyses obtained on NW cross-sectional focus ion beam (FIB) lamellae show that substrate orientation induces a variation in core morphologies. Moreover, we find that atomic steps morphologies are the main cause for strain field modifications in the shell crystal. To reveal, in a sub-nanometer scale, the influence of the core morphology on the electronic band alignment of the 1D core-shell heterostructure, we perform a detailed scanning transmission electron microscopy (STEM) study. Furthermore, we have developed an iterative methodology based on electron energy-loss spectroscopy (EELS) and used it to map the band-gap evolution in the complex core-shell structure with high spatial accuracy (including a sub-nanometer pixel size map). The obtained results highlight the nanoscale electronic band-gap modulation due to the high strain at the core and shell interface region, including a switch from direct to indirect band gap in the ZnTe interface region. A similar strain-induced switch was reported in wurtzite GaAs[53], however, here we report that such strain-induced effect is observed with sub-nanometer resolution precision in a confined region of a heterostructure. With the present study, we show the importance of detailed atomic-scale crystal evaluation for controlling and tuning optoelectronic properties in horizontal 1-D arrays, which is of great relevance in the fields related to electronics, photonics, quantum applica-tions, and photovoltaics among others.

**Crystalline structure.** Most II-IV binary compounds crystallize in two main polytypes: cubic Zinc Blende (ZB) and hexagonal Wurtzite (WZ)[54,55], which differ in their respective stacking sequences when projected through the <110>$_{ZB}$ or <11$\bar{2}$0>$_{WZ}$ zone axes but present an identical projected appearance when observed through the <112>$_{ZB}$ or <1$\bar{1}$00>$_{WZ}$ direction. For this reason, we

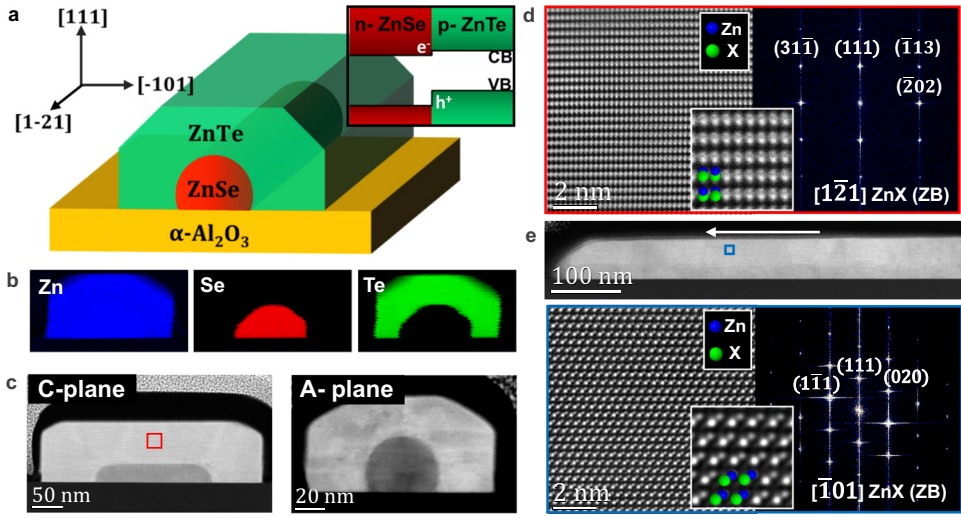

**Fig. 1 Structure, composition and crystallographic orientations of sapphire-supported core-shell NWs. a** Schematics of NW configuration, ZnX crystal directions (X = Se/Te) and core-shell band alignment. **b** EELS-based elemental maps for Zn (blue), Te (green) and Se (red). **c** Low-magnification HAADF-STEM micrograph of NW transversal cross-sections on A-plane and C-plane sapphire. **d** Atomic-resolution HAADF-STEM detail and its corresponding power spectrum (FFT) of the transversal cross-section shown in (**c**). The inset reveals the atomic arrangement in more detail for precise chemical identification through the atomic-number contrast (Z-contrast). **e** Low-magnification HAADF STEM micrograph of a longitudinal cut on a NW grown on C-plane sapphire, and higher magnification detail (inset) with its corresponding FFT of the area highlighted with a blue square.

used Aberration Corrected High Angle Annular Dark Field (AC-HAADF) STEM imaging in two perpendicular directions of the studied NWs for determining and evaluating the crystal structure and quality of the core-shell structures grown on two different substrates (A- and C-plane sapphire). FIB lamellae were obtained by longitudinally cutting a single NW or transversally cutting several NWs of the array. We evaluated the elemental distribution by acquiring EELS-based composition maps for Zn, Te, and Se (Fig. 1b). These maps reveal a sharp interface between core and shell with no anionic diffusion at the nanometer scale. To ascertain the crystal structure and orientation, an atomic resolution AC-HAADF STEM micrograph was obtained in the squared region (Fig. 1d), allowing us to identify the [1$\bar{2}$1] crystal direction as the growth direction. From an additional longitudinal cross-section of the NW shown in Fig. 1e, the cubic ZB phase imaged through the [$\bar{1}$01] zone axis can be unequivocally identified. We determine that the overall growth direction of the NW is a <112> direction (for both A- and C-plane substrates) while the growth front corresponds to a <111> direction. This second structural projection also shows the lack of stacking defects along the length of the nanowire, which we find for both substrates used in this study.

With closer inspection of the high magnification micrographs (Fig. 1d, e), the Z-contrast arising in HAADF STEM imaging allows for atomic column identification. The nanowires grow following a B-polar (anion-polar) <111> direction, which is the typical growth direction observed for most III-V and II-VI semiconductor binary compound materials[24,26].

**Strain analyses**. ZnTe and ZnSe present lattice constants of 6.09 Å[56] and 5.67 Å[57], respectively, which correspond to a lattice mismatch of 7.4%. This high mismatch creates strain fields that can produce misfit dislocations at the heterostructure interface due to the high accumulation of elastic energy. As mentioned above, strain fields and dislocations are critical since they play a role in the (opto)electronic behavior of semiconductor materials[58]. Furthermore, complex geometries in highly mismatched planar configurations are responsible for the formation of intricate lattice deformations.

In this regard, Geometric Phase Analysis (GPA) has been applied as an analytical technique to obtain atomic scale information by examining the dilatation and rotation of crystal planes, allowing for high spatial resolution in strain analyses[59,60]. While the vapor-solid (VS) growth of the shell leads to a fixed geometry formed by two lateral {$\bar{2}$02} non-polar planes, a large top (111)A plane and two minor {$\bar{1}$13}A planes, different core morphologies for each orientation can be observed in the case of the VLS growth of the core (Figs. 1a–c and 2a, b). The analyses presented in Fig. 2 were performed with atomic resolution micrographs obtained from transversal cross-sections of VLS-grown samples on both A- and C-plane sapphire. On C-plane sapphire, core morphologies tend to present highly faceted morphologies that can range from planar belts (Fig. 2a) with a predominant (111) top facet to faceted structures with the development of lateral {$\bar{2}$02} facets and some minor {$\bar{1}$13} corners, resembling the faceting of the ZnTe shell (Supplementary Fig. 2). In contrast, on A-plane sapphire, core faceting is barely present and core morphologies become either truncated cylinders in the most extreme case (Fig. 2b) or elliptic cylinders presenting different curvatures and some residual top facets. These core morphologies with rounded cross-section present a clear suppression of the lateral {$\bar{2}$02} facets and contact angles with the substrate are higher than 90° (Supplementary Fig. 2). We attribute the formation of different morphologies to the different surface energies of C and A sapphire planes, which modify the contact angle of the catalyst and thus influence the final morphology of the deposited material.

Fast Fourier Transforms (FFTs) of the HAADF STEM images show a clear difference in ZnTe crystal planes for NWs on A-plane and C-plane sapphire. In every crystalline reflection we expect to see two distinct spots, arising from an epitaxial core-shell structure presenting the same crystal orientation but with different lattice constants that are indicative of the different materials, as clearly observed in Fig. 2c. The inner spots correspond to the ZnTe lattice and the outer spots to the ZnSe lattice. However, in A-plane sapphire we found a triple spot where the outer spots correspond to the ZnSe core and the inner ZnTe shell spots split into two

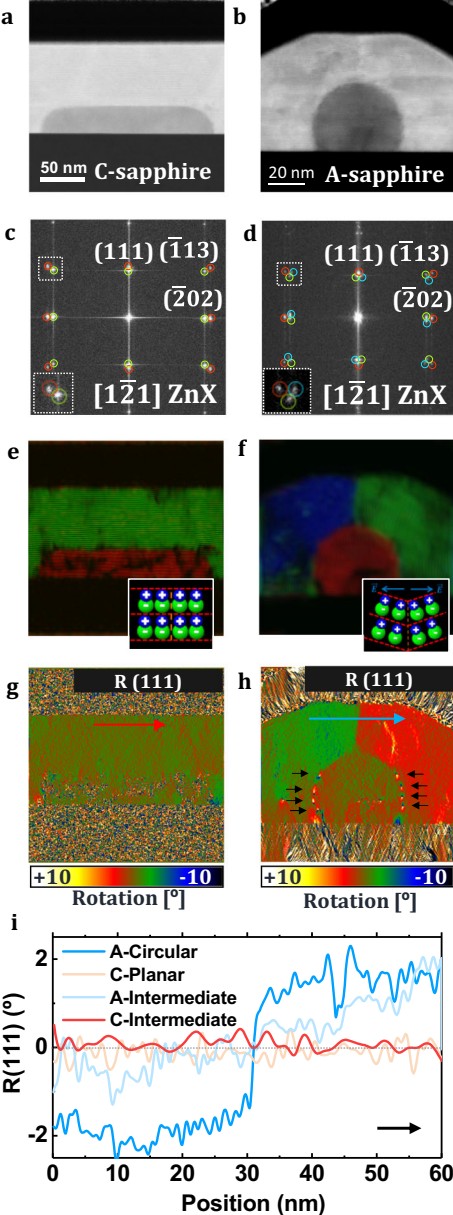

**Fig. 2 Study of the shell rotation.** Core-shell ZnSe@ZnTe NWs on C-plane sapphire (left, **a**, **c**, **e**, **g**) and A-plane sapphire NWs (right, **b**, **d**, **f**, **h**). **a**, **b** Low-magnification atomically resolved HAADF STEM image showing a representative NW cross-section. **c**, **d** Plane-indexed FFTs corresponding to (**a**) and (**b**), highlighting ZnSe core planes (red) and ZnTe shell planes (green/blue to differentiate orientations) in these structures. **e**, **f** Frequency-filtered RGB structural maps (IFFT) with color codes as indicated in (**c**) and (**d**). The insets schematically show dumbbell orientations in the central part of the ZnTe shell. **g**, **h** GPA (111) plane rotation (R) maps relative to the NW core planes, which are taken as reference. Color bars represent the geometrical phase analysis calculated (111) planes rotation in degrees (°). Black arrows in (**h**) indicate dislocation positions. **i** (111) plane rotation profiles across the center of the NW shell (following the directions indicated by the red and blue arrows in (**g**) and (**h**), respectively) for relevant core morphologies in both C-plane and A-plane substrates.

reflections that differ by 4° (Fig. 2d). Filtering each distinct set of planes (Fig. 2e, f) reveals the existence of four well-defined structural sections in A-plane sapphire NW shells, while a simple monocrystalline shell structure is observed for C-plane sapphire.

The four A-plane sapphire NW shell sections differ in the plane rotation but retain the same interplanar distance, matching that of relaxed ZnTe (Supplementary Fig. 4). More information can be ascertained on this anomalous rotation phenomenon when analyzing the rotation maps obtained by GPA. In the rotation maps of (111) planes we observe that part of the strain is released by the creation of misfit dislocations, which are especially well visible in Fig. 2h. Atomic resolution details of the misfit dislocations in both NWs are shown in Supplementary Figs. 11 and 12. However, we also find elastic deformation of the shell in the form of a plane rotation variation on both sides of the NW which merge forming a sharp ±2° bent boundary. In the other nanowires grown in A-plane sapphire, which do not show a perfectly truncated circular cross-section, some bending is also found (Supplementary Fig. 3). However, the maximum rotation achieved there lies in the range of ±1–1.5°, and instead of featuring sharp boundaries, the change in rotation is gradual. This phenomenon is not observed in the flat or faceted nanowires grown on C-plane sapphire, where we could not measure any plane rotation, neither in the flat nor in the partially faceted NWs, as summarized in Fig. 2i and detailed in Supplementary Figs. 3e–h and 5.

The nature of strain relaxation in the core-shell heterojunction has been determined through the experimental measurement of plane dilatations in two orthogonal directions, [111] and [20$\bar{2}$], to obtain the strain tensor components ($\varepsilon_{ij}$) with respect to the core, in which the relaxed ZnSe lattice constant is measured as reference. The high $Z^2$ average difference between ZnSe and ZnTe in comparison to sapphire ($Al_2O_3$) saturates the contrast and makes sapphire invisible in HAADF STEM images, the technique of choice for our GPA analyses as it provides low environmental noise. Therefore, we could not simultaneously image both materials in HAADF STEM. However, by using HRTEM images, we measured the lattice constants of the core in comparison to the substrate in several NWs, and we found that the ZnSe core lattice parameters matched with its relaxed value, so we could take this value as a reference in our GPA maps and measure the distortions caused in the shell with respect to the core reference. Details on these calculations can be found in Supplementary Note 3. The overall dilatation of ZnTe planes with respect to the ZnSe lattice are compatible with a relaxed shell within the level of experimental accuracy. This result, in addition to the misfit dislocation array revealed in the rotational maps and the direct atomic resolution HAADF STEM images (Fig. 2g, h and in more detail in Supplementary Figs. 11 and 12), show that the core-shell heterostructure releases part of the strain plastically by the creation of misfit dislocations and then achieves relaxation by releasing the remaining strain by compressing the ZnTe lattice in the 3–7 nm region closer to the interface (Supplementary Figs. 4 and 5). The same behavior is observed for the NWs grown on both A- and C-plane sapphire substrates, so they only differ in the appearance of the different sections in the NW formed by plane bending.

We now elucidate the origin of this plane rotation. Considering a vertical interface between core and shell, a plane rotation β in the shell planes modifies the effective interplanar distance of the shell planes in contact with the core planes as $d_{shell}^{eff} = d_{shell}/\cos\beta > d_{shell}$ (Supplementary Fig. 6). This leads to tilted epitaxy when $d_{core} > d_{shell}$ since the effective mismatch between the two structures is reduced. The result is a more stable configuration in terms of elastic energy[61]. Nonetheless, when $d_{core} < d_{shell}$, the tilting of shell planes would instead increase the effective mismatch, and thus, the elastic energy of the system. Therefore, the origin of plane bending is related to the specific morphology of the core. To explore this hypothesis, atomic models reproducing the A-plane sapphire NWs were created, their corresponding HAADF STEM images were simulated[62,63], and GPA rotation maps were obtained. Exploiting

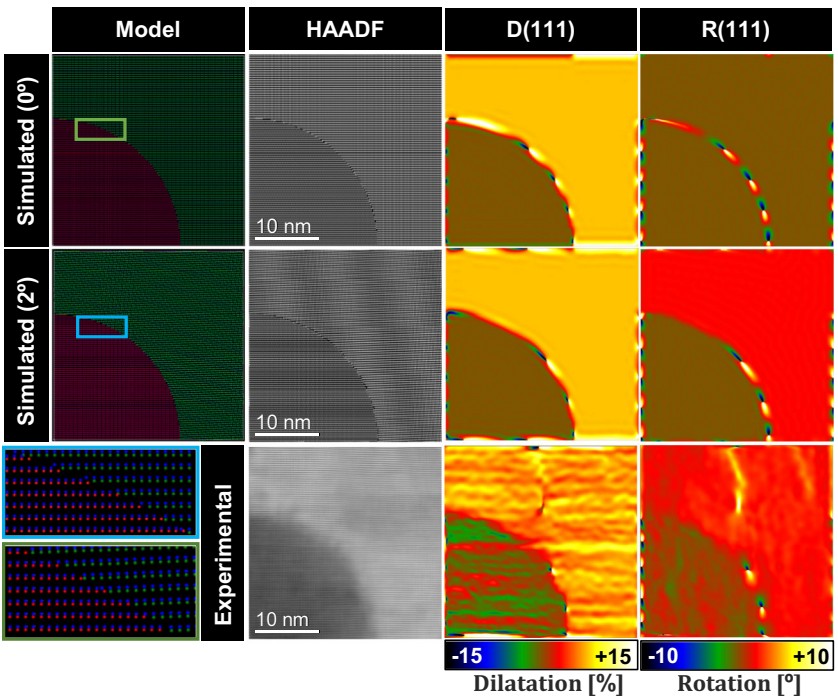

**Fig. 3 Study of the shell relaxation mechanism.** (Two-upper-rows) (Model column) Atomic modeling of a NW top-right section reproducing the experimental morphology (ZnTe presenting a 2° rotation around the1–21 axis), along with a hypothetical configuration with ZnTe planes completely parallel to ZnSe ones (0°). (HAADF Column) HAADF-STEM micrograph simulations of both configurations along with GPA dilatation (D) and rotational (R) maps for the (111) structure planes of the simulated models in comparison with the equivalent experimental analysis in the same NW section (bottom-row). The left color bar represents the geometrical phase analysis calculated (111) planes percentage dilatation (%), while the right color bar represents the geometrical phase analysis calculated (111) planes rotation in degrees (°).

the symmetry of the structure, only the top-right section of the nanowire was modeled since the results can be extrapolated to the other half of the NW. Different shell rotations (from 0° to 5° with steps of 1°) were modeled (see Supplementary Figs. 7–9).

We found that the core and shell structures present the best matching for a 2° rotation, resulting in a minimization of the elastic energy accumulation at the interface, in good agreement with experimental results (Fig. 3). This is reflected by the lower number of misfit dislocations at the core-shell interface, as well as higher matching with planes that are tangential to the interface. More details on the strain relaxation mechanisms together with a shell rotation study can be found in Supplementary Note 5.

Due to the polar nature of the bonds in II-VI and III-V semiconductors, the dumbbell orientation plays a key role in determining the materials' properties[25]. In fact, the presence of polar defects such as anti-phase boundaries (APBs) or inversion domains has been reported as detrimental to the electronic behavior of binary semiconductors, since they create local electric fields and potential barriers that critically decrease their conductance[24,26]. In the pristine ZnTe structure, the vertical <111> direction exhibits a polarity due to the dumbbell (dipole) orientation while the horizontal <20$\bar{2}$> direction remains non-polar. However, the presence of plane bending around the growth axis direction in A-plane NWs (Fig. 2f and corresponding inset) creates a configuration similar to APBs or para-twins as, at the center of the NW shell, the Zn extremes of the dipoles are facing each other in the low-angle boundary, breaking the horizontal polar neutrality across the NW shell vertical axis and leading to the possible creation of an electrostatic potential barrier at the boundary, as suggested elsewhere[24,26]. Therefore, controlling core morphologies is essential to optimize the shell epitaxy and avoid morphology induced low angle polar boundaries.

**Band-gap measurements through Valence EELS**. Low-loss electron energy-loss spectra contain information about the allowed optical and electronic transitions of the material, including collective excitations such as phonons and plasmons, as well as single-electron transitions[64–66]. In the low-energy region of the loss spectra, valence EELS (VEELS, 0–10 eV) reflects the joint density of states (JDOS), whose dependence on the energy losses is $(E-E_g)^{1/2}$ in a direct band-gap material, and $(E-E_g)^{3/2}$ in an indirect band-gap material[67], where $E$ is the energy lost by an electron inelastically scattered by the specimen, and $E_g$ is the local band-gap energy at the electron-sample interaction site. Consequently, VEELS is a useful technique for band-gap evaluation with the advantage of offering high spatial resolution. However, the complexities of the required data analysis, in addition to the presence of parasitic effects such as Cherenkov radiation losses, delocalization effects and signatures from excitons and surface-state excitations, limit the applicability of this technique for band-gap determination. These limitations are especially dramatic in semiconductor materials that present a strong Cherenkov contribution, often occurring in those with high refractive indices[48,49]. As described earlier, the band gap of radial p-n junctions is of special interest because it determines the main emission lines. Particularly, ZnTe and ZnSe present band gaps lying in the visible range (2.26 eV[47,68] and 2.67 eV[69], respectively, both at room temperature). These band-gap energies can be susceptible to modulations due to the atomic arrangements and strain distribution at the core/shell interfaces. We, therefore, evaluated the band-gap properties of the ZnSe@ZnTe system despite the experimental difficulties. Moreover, in spite of possible additional contributions such as surface modes, it was relatively straightforward to determine a gap value of 8.8 eV for the sapphire substrate[70], providing a strong incentive to establish

                                                    

a methodology to do the same with the lower gap materials that constitute the NW.

In order to overcome these limitations, one can perform VEELS simulations to compute the spatial and energetic dependence of the parasitic signal and compare it to the total signal, in which the local DOS information lies[71,72]. With this approach, we can subtract the parasitic contributions in the experimental EELS data and perform meaningful fittings to extract physically representative information. The details on the applied methodology are described next, while further details can be found in Supplementary Note 7.

We employed several approaches in order to distinguish the different regions of interest across the spatially resolved EELS data, or spectral image, and ensure an adaptative response of the fitting routine. For the methodology developed here, we used the main plasmon peaks of both ZnTe (centered at 15.3 eV) and ZnSe (centered at 16.9 eV), as well as the one of $Al_2O_3$ (centered at 25.4 eV) to map the local distribution of these materials, and threshold them to ensure they coincide with the domains observed in the spectral image, avoiding the effects of plasmon delocalization. Unsupervised machine learning has been used by means of Multivariate Statistical Analysis (MSA)[73,74], to obtain component spatial maps of the interface pixels between ZnTe and ZnSe. The spatial distribution of the components is in agreement with the medium angle ADF (MAADF) STEM image presented in Fig. 5b, which highlights the regions containing strain accumulation (e.g., near interfaces and volumetric defects)[75,76]. The presented results rely on the ability of the fitting routine to capture the spatial distribution of both these components and the plasmonic signatures of every material. These routines show the potential of extending this idea further, for instance, when a material composition modulation is expected (e.g., elemental diffusion or alloying). In this latter case, simultaneous core-loss EELS and VEELS (in a dual-EELS system) or simultaneous EDX and VEELS could be useful to correlate the exact composition (obtained by core-loss EELS or EDX chemical maps) of the material in every pixel with the electronic structure we are interested in (collecting the inelastically scattered electrons, i.e., VEELS). Therefore, a precise material identification based on core-loss compositional quantification could be done, allowing for more accurate VEELS simulations to eliminate parasitic contributions in the experimental EELS signal. In addition, machine and deep learning could improve the methodology with routines for automatic image segmentation[77,78].

The ability to spatially identify the pixels corresponding to each of the materials of interest allows the subsequent tailored-data processing assisted by STEM-VEELS simulations of the device. These simulations are based on classical electrodynamics, with the materials described by their frequency-dependent, local dielectric functions. The energy-loss probability is directly separated as the sum of a bulk contribution, which is independent of geometry and proportional to the path length traveled inside each material, and a surface term determined by the interface morphology[70]. We perform both retarded and non-retarded (i.e., assuming an infinite speed of light) calculations and compare them to assess the role of Cherenkov radiation emitted by the fast electrons in their interaction with the bulk materials. Given the translational invariance of our system along the electron-beam direction, the bulk contribution ($\Gamma_{Bulk}^R$) is the same as for an electron moving in an infinitely extended material, while the interface corrections ($\Gamma_{Surf}$) are determined by solving Maxwell's equations with the boundary-element method[71,72,79]. We find converged results when the interfaces are parametrized with points separated by 0.3–0.5 nm, which incidentally corresponds to the spatial resolution in our experimental spectral images. Although this precision does not

reach atomic resolution, it allows us to convolute the atomistic effect of dislocations. Quantum confinement effects are safely ignored because of the large length scale (~100 nm) involved in the nanowire structure. Our simulations assume perfect crystals, although complementary simulations demonstrate the robustness of the method with respect to strain and a band-gap change smaller than ±1 eV (Supplementary Notes 7 and 8).

Based on the information provided by the simulated retarded and non-retarded spectral images, we correct the experimental data to subtract parasitic contributions associated with Cherenkov radiation and excitation of surface modes. By geometrically correlating the simulated device shape with the experimental one, a correction function is pixelwise applied to the experimental spectral image to only retain the experimental loss probability corresponding to the bulk excitations of interest. The correction function, which depends both on position (pixel) and on energy (spectrum channel), is the ratio between the non-retarded bulk loss ($\Gamma_{Bulk}^{NR}$) and the total retarded loss ($\delta(r\prime, \omega) = \frac{\Gamma_{Bulk}^{NR}}{\Gamma_{Total}}$, where $\Gamma_{Total} = \Gamma_{Bulk}^R + \Gamma_{Surf}$). Each of the terms involved in the correction factor depends linearly on the lamella thickness, making the process robust against thickness modulations or inhomogeneities. In our case, the lamella thickness was estimated to be 45 nm thick and homogenous. The entire workflow of data cleaning and processing assisted by the simulations is described in Fig. 4. The detailed algorithmics behind these corrections are discussed in Supplementary Notes 7 and 8).

The so-cleaned experimental data (free of interfering Cherenkov and surface-related signals) are ready to perform a meaningful analysis. We proceed by implementing adaptive and material-dependent curve fittings in a pixelwise fashion to determine the band-gap structure (direct or indirect) that better represents the nature of the observed spectra[67]. Our method is not limited to solely determining the band-gap value; it can also disclose its nature by discriminating between direct and indirect band-gap materials in complex heterostructures, which is achieved by minimizing a reliability factor $r^2$ when fitting a model function to the experimentally measured gap onset (see more details in Supplementary Notes 7 and 8, and particularly in Supplementary Figs. 13–18). This process allows us to extract a physically reliable band-gap map (Fig. 5c), which we have split into two independent maps that highlight those pixels that represent either a higher probability of fitting a direct band structure (Fig. 5d), or an indirect band structure (Fig. 5e).

The band-gap values remain close to the ones expected for the relaxed materials (i.e., the following average band-gap values from the full extension of each material and region with the corresponding standard error of the mean: "bulk" ZnTe pixels = 2.3162 ± 0.0012 eV; 'bulk' ZnSe = 2.7733 ± 0.0012 eV; α-$Al_2O_3$ = 8.271 ± 0.016 eV). However, we observe a significant band-gap energy reduction (statistical significance tests are provided in Supplementary Note 6) in regions where strain is accumulated, as inferred from the MAADF STEM images, with average band-gap values from ZnTe's interface pixels: 2.28 ± 0.01 eV; and from ZnSe interface pixels: 2.755 ± 0.011 eV. The strained regions in the MAADF STEM image matching with the locations where the band gap shows lower energy values indicates the modulation of the heterojunction at the active area, the interface between both materials (Fig. 5f). The sharpness of the transition of the type II recombination expected from the p-n junction is justified by the high-quality epitaxy and sharp interface and the consequent possibility to adapt the fitting whether the pixels correspond to ZnTe or ZnSe, indistinctively.

In the unstrained ZnTe and ZnSe areas, according to our GPA analyses and MAADF-STEM images, our VEELS measurements reveal a direct band-gap type in those ZnTe and ZnSe areas that are shown as relaxed in our GPA analyses and MAADF STEM

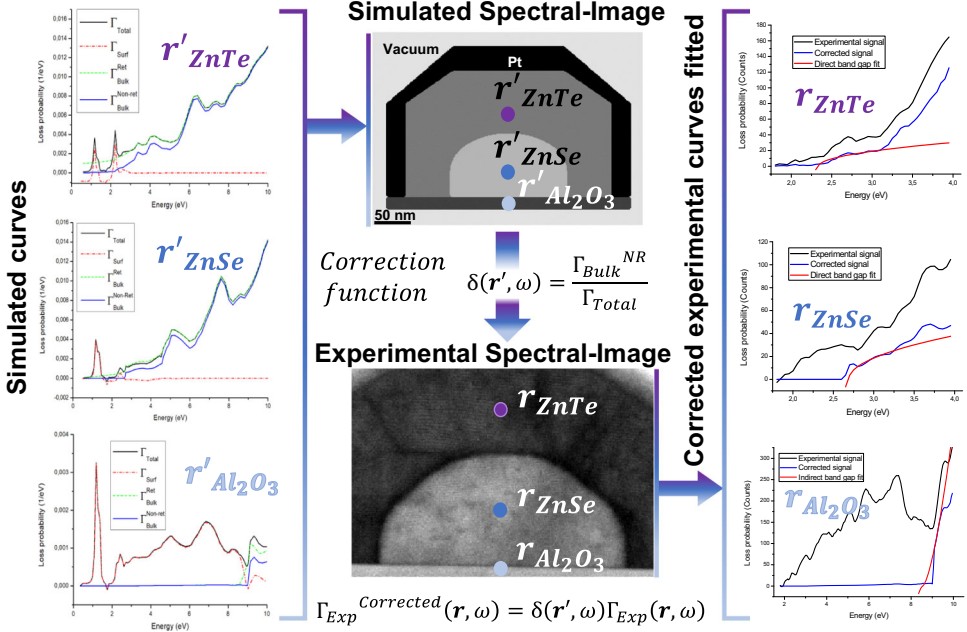

**Fig. 4 Workflow of the developed method for correcting the measured low-loss Electron Energy-Loss Spectroscopy (EELS) data affected by Cherenkov radiation and surface modes.** Classical electrodynamics simulations are used to produce a spectral image of both the non-retarded and retarded loss probabilities, which are by construction decomposed into bulk and surface contributions. Next, a correction function that compares the simulated non-retarded and retarded losses is used to map the effect of parasitic radiation and to clean the experimental spectral image. Finally, the corrected experimental signal is used to fit either a direct or indirect band-gap structure depending on the material and spatial position, and to extract a band-gap value from such fit.

images, while a clear indirect band-gap behavior is found for the relaxed α-Al2O3 substrate (Supplementary Fig. 14)[80]. Interestingly, in the 3 nm-thick ZnTe shell region close to the core interface, the fitting obtained with a direct band-gap function produces noisy results with both lower goodness of fit and higher error. Thus, we tried the fitting for the VEELS corresponding pixels in this region with an indirect band-gap function. The indirect fitting offers a better performance in terms of meaningful resulting physical information and in goodness of fit, significantly surpassing the direct fitting with a confidence level higher than 99.9995% (see "Significance tests" in the Supplementary Information). Examining this area in the GPA calculations and MAADF-STEM contrast images suggests that a high degree elastic strain remains there. The details on this evaluation can be found in Supplementary Note 8. This difference in fitting performance and obtained average band-gap values per region suggests a band type transition from direct to indirect of ZnTe under compressive strain (see Fig. 5f and further details in Supplementary Note 8) and supports what was predicted theoretically by DFT calculations in the literature[81]. From this theoretical study, we conclude that the direct interband transition at the Γ point would change from valence Γ to conduction Z (indirect), due to a decrease in the conduction band minimum at this Z point. With its high local accuracy (0.9 nm pixel size) and the tolerance toward Cherenkov radiation, our measurements constitute, to the best of our knowledge, the first evidence in which high sensitivity band-gap structure characteristics have been reported experimentally.

The guided growth of core-shell NW horizontal arrays involves strong epitaxial interactions between materials that directly affect and modulate the electronic behavior of the effective heterojunction.

An accurate sub-nanometer scale study of the local strain-induced electronic band modulation has been performed in the present work. Substrate orientation and an appropriate substrate-

catalyst wetting control affect the core curvature and final cross-section morphology during horizontally guided NW growth. Core curvature has been shown to be the cause of induced plane rotations in the shell crystal structure. These rotations tune the overall electronic band structure of the system by a phenomenon directly arising from atomic steps edging nanowire cores.

A further developed VEELS methodology has allowed us to obtain a full system band-gap map with sub-nanometer scale accuracy that offers a clear experimental fingerprint of a band gap energy decrease together with a direct to indirect band-gap transition, induced by the epitaxial strain accumulated at the 3 nm-thick ZnTe shell region close to the core interface. Our study provides accurate and interesting tools for the electronic characterization of nanostructured semiconductors in the framework of electron microscopy that are crucial for understanding and designing proper complex electronic, photonic and quantum devices, especially in the recently developed field of horizontally grown NW networks and circuits.

## Methods

**(S)TEM imaging and spectroscopy.** The transversal and longitudinal cross sections were prepared by a Focused Ion Beam (FIB) FEI Helios 650 and HAADF imaging was performed in a FEI TITAN Low-Base operated at 300 kV. Lamellae were prepared using Ga-ion FIB operated at voltages ranging between 30 kV in the first stages to 1 kV in the last steps. This procedure reduced FIB-induced damage and Ga implantation to a minimum, also resulting in an amorphized layer thickness below 1 nm, which had a negligible impact on the imaging and spectroscopy analyses[82,83]. EELS elemental maps were obtained in a FEI TECNAI F20 TEM microscope operated at 200 kV. VEELS measurements were performed in a monochromated Nion UltraSTEM[TM] 100MC "HERMES" corrected to the 5th order and operated at 60 kV. The probe convergence semi-angle was 31mrad, with the monochromator slit adjusted to provide an energy resolution of approximately 150 meV (as estimated from the full-width at half maximum of the zero-loss peak), offering sufficient resolution for the gap values expected while maintaining high probe currents. Given the dispersion used and the detector's point spread function,

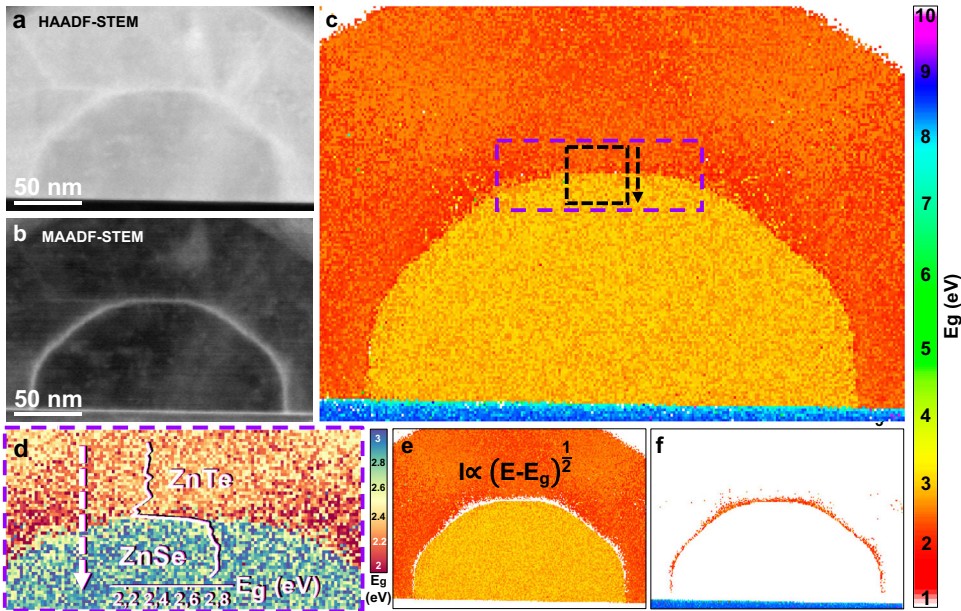

**Fig. 5 Measured band-gap maps. a** HAADF-STEM and **b** MAADF-STEM micrographs of a ZnSe-ZnTe core-shell nanowire grown on C-plane sapphire to which the band-gap mapping methodology has been applied. **c** VEELS-band-gap map of the system. The dotted purple square indicates the zoomed-in area displayed in (**d**), in which the 2–3 eV scale allows for a better qualitative representation of the theoretically predicted decrease in the band-gap at the strained ZnSe-ZnTe interface. This tendency is quantitatively observed in the black curve superimposed in (**d**), which has been horizontally averaged over the pixels included in the black dotted box and along the black arrow direction in (**c**). VEELS-band-gap map decomposition into the regions in which we observe either (**e**) a direct band structure (all ZnSe pixels and ZnTe "bulk" pixels), or (**f**) an indirect band-gap (ZnTe shell pixels at the core interface, and sapphire pixels). The pixels corresponding to the metallic polycrystalline Pt protective layer have been manually set to 0 eV (metallic behavior). Notice that the color bar represents the band-gap energy in eV.

the zero loss peak (ZLP) was Point-spread-function-limited at ~150 meV wide. High-angle and medium angle annular field image detectors' angular ranges are 95–190 mrad and 55–95 mrad, respectively. EELS was acquired using a Gatan Enfinium ERS spectrometer, with an entrance aperture semi-angle of 44 mrad. The atomic resolution HAADF STEM images shown in Fig. 1d, e have been filtered by using a Wiener filter plus a beam deconvolution in STEM-CELL software[62,63].

**(S)TEM data analyses**. Analyses were carried out using Gatan DigitalMicrograph software. Geometric Phase Analysis is performed using GPA v4.10 plugin within DigitalMicrograph. Analyses were performed on (111) and (20-2) core-shell reflections using a cosine mask type.

**Atomic modeling and HAADF simulation**. Atomic models were generated with Rhodius software from Universidad de Cadiz[84,85], considering the experimental cross-section dimensions and 10 nm thick supercell. HAADF simulations were performed in STEM-CELL software[62,63] using a linear approximation by employing 300 kV energy, C ap 15 μm, defocus 0 nm.

**VEELS modeling**. Boundary-Element Method simulations were performed in order to calculate the contribution of interface terms to EEL spectral-images, using a two-dimensional implementation that takes advantage of translational invariance in the structures[71,72,79].

## Data availability
All raw and processed data (including images and spectra) that support the findings of this study have been deposited in GitHub with the primary accession codes "mbotifollmoral/ZnTe-ZnSe_Strain_Modulated_BG_Mapping (https://github.com/mbotifollmoral/ZnTe-ZnSe_Strain_Modulated_BG_Mapping.git)". In addition, raw data were generated at the Catalan Institute of Nanoscience and Nanotechnology (ICN2), the SuperSTEM Laboratory and the LMA-INMA large-scale facilities.

## Code availability
The codes and algorithms used to obtain the results of this study have been deposited in GitHub with the primary accession codes "mbotifollmoral/ZnTe-ZnSe_Strain_Modulated_BG_Mapping (https://github.com/mbotifollmoral/ZnTe-ZnSe_Strain_Modulated_BG_Mapping.git)".

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

## Acknowledgements

J.A., S.M.-S., M.B., and M.C.S. were supported by the Severo Ochoa program from Spanish MCIN/AEI /10.13039/501100011033/ (Grant No. SEV-2017-0706) and are funded by the CERCA Program / Generalitat de Catalunya. Part of the present work has been performed in the framework of Universitat Autònoma de Barcelona Materials Science Ph.D. program. J.A., S.M.-S., M.B., and M.C.S. acknowledge funding from Generalitat de Catalunya 2017SGR327. S.M.S. acknowledges funding from ICN2 SO EXEM project. The HAADF-STEM microscopy was conducted in the Laboratorio de Microscopias Avanzadas at Instituto de Nanociencia de Aragon-Universidad de Zaragoza. Additional imaging and electron energy loss spectroscopy was carried out at SuperSTEM, the UK National Research Facility for Advanced Electron Microscopy, supported by the Engineering and Physical Sciences Research Council (EPSRC). Authors acknowledge the LMA-INMA for offering access to their instruments and expertise. We acknowledge support from CSIC Research Platform on Quantum Technologies PTI-001 (QTEP). M.B. acknowledges support from SUR Generalitat de Catalunya and the EU Social Fund; project ref. 2020 FI 00103. C.K. acknowledges funding from InPhINIT Scholarships LaCaixa Foundation. M.C.S. have received funding from the European Union's Horizon 2020 research and innovation program under the Marie Sklodowska-Curie grant agreement No 754510 (PROBIST). J.G.d.A. has been supported by European Research Council (Advanced Grant 789104-eNANO). V.D.G. and J.G.d.A have received funding from Spanish MCINN (PID2020-112625GB-I00 and CEX2019-000910-S), Fundació Cellex, Fundació Mir-Puig, and Generalitat de Catalunya (CERCA, AGAUR). E.J. acknowledges support from the Israel Science Foundation (Grant No. 2444/19). J.A., S.M.-S., M.B. and M.C.S. were supported by MCIN with funding from European Union NextGenerationEU (PRTR-C17.I1), by Generalitat de Catalunya and by "ERDF A way of making Europe", by the "European Union".

## Author contributions

S.M.-S., M.B., and J.A. conceived and initiated the project. J.A. supervised the project and led the collaboration efforts. S.M.-S., M.B., and J.A. designed the experiments. E.O. and E.J. took care of the samples growth. S.M.-S., M.B., C.K., C.B., M.C.S., Q.R., and J.A. obtained and analyzed the TEM, STEM, cross-sectional STEM measurements and EELS spectra. M.B., V.D.G., and F.J.G.A performed the theoretical analyses and models. S.M.-S., M.B., and J.A. wrote the paper. All authors discussed the results and commented on the manuscript.

## Competing interests

The authors declare no competing interests.
