## [Peer Review File · Nature Communications]

This manuscript has been previously reviewed at another journal that is not operating a transparent peer review scheme. This document only contains reviewer comments and rebuttal letters for versions considered at *Nature Communications*.

REVIEWER COMMENTS

Reviewer #1 (Remarks to the Author):

The manuscript addresses the structural interplay at the interface between core and shell of planar arrays of core-shell semiconductor nanowires where the strain is not as easily released as in free standing nanowire. The manuscript presents impressive experimental data and careful data analyses and modeling to extract detailed information about the spatial variation in strain at the interface.

The structural rearrangement at the core-shell interface and spatial variation in strain affect the band gap and also causes a local transition from direct to indirect band gap as an effect of compressive strain on the sub.nanometer scale.

The methodology is sound, the work is of high quality and well described.

I have one question about the data presented in Figure 1c. There is a contrast variation in the shell where thin lines of brighter contrast are seen stretching from the corners of the core to the surface . A comment on this would be welcome.

The observations are of high significance to the field. I recommend this manuscript to be accepted for publication.

Reviewer #2 (Remarks to the Author):

Papier : “Sub-nanometer mapping of strain-induced band structure variations in planar nanowire core-shell heterostructures” by Sara Martí-Sánchez et.al. , present interesting attempt of mapping of local bandgap in strained laterally grown nanowire. The authors claim to link the experimentally determine spatially resolved bandgap with local strain in thin cross-sections of such heterostructure .

The general remarks:

1. The detail of procedure/method of determination of local bandgap should be shifted in condensed form from supplement to main paper. This part is critical in support of the one of the thesis of the article: “Our analysis reveals a decrease in band gap energy decrease at highly strained core-shell interfacial regions, along with a switch from direct to indirect band gap, as inferred by fitting to their corresponding spectral functions”.
2. All measurements are performed on thin FIB cross-section. However, the impact of lamella thickness (which can be mapped by EELS), on both, the local strain and bandgap, are not taken into account or discussed in analysis/EELS model.
3. What is the impact of interface misfit dislocations that can act as low bandgap centers? What about influence of surface states related to FIB radiation damage/amorphisation of lamella surfaces?

Other remarks:

4. Whether the use of the term "crystallinity, which is gaining more and more popularity is appropriate here for defective but monocrystalline heterostructure? Originally, this term described the ratio of the crystalline phase to the amorphous phase in materials such as polymers determined by X-ray diffraction. Authors have details information about structure from TEM. So more precision descriptors can be used.
5. At Figure 1a the indication of crystallographic direction should, by convention be [-101],

not [-202]. What material is present in the black area between FIB platinum and NW in fig 1c?

6. Authors claim that sapphire orientations determine orientation of laterally grown NW. Nevertheless, high-resolution images of Sapphire/NW interface are not given. The sapphire substrate looks like masked in figure 1c, 2a,b. As well, Bragg peaks from sapphire are not present on FFT fig 2 c,d allowing confirmation of epitaxial relations on NW and substrate. The low magnification image of substrate/NW can only be seen in figure SI13. Authors should include high-resolution images of sapphire/NW boundaries to prove that this interface is semi coherent and force orientation of NW crystal lattice.

7. The main finding of the paper is related to interface region between ZnTe shell and ZnSe core. Geometric models and GPA analysis suggest the presence of misfit dislocations at these interfaces. What is the nature of these dislocations (Lomer, 60 degree etc..) Authors present the high clarity filtered, atomic resolution STEM images of “atoms arrangement “in perfect part of NW (fig 1d,e). The example of high resolution image with interface area containing misfit dislocation allowing a determination of projected components of Burger’s vector of these dislocations should be given.

8. The discussion about, the impact of the sapphire on strain distribution in NW determined by GPA should be added. GPA measurements give relative values of lattice distortion to the reference area taken in the center of ZnSe core. Are the NWs fully relaxed in relation to sapphire substrate?

9. It seems that some references in text to the figures in supplement are not valid. For example, there are two figures indicted as SI3 in supplement.

10. The sentence “Examining this area in the GPA calculations and MAADF-STEM contrast images suggests that a high degree elastic strain remains there.” is used twice in the same paragraph.

Reviewer #3 (Remarks to the Author):

This is a high-level paper, which is well written, presenting a timely topic. I would support publication of this work, but the balance and the accessibility of the paper may be optimized: a lot of text has been spent on Figures 1 and 2 (up to page 12), while these present a rather straightforward characterization of the material system. Maybe this first part can be shortened. I find Figures 3 and 4 the most interesting, but also difficult to follow. A background on TEM is needed, and there are many references to the supplementary information. Maybe more explanations can be included in the main text in this second part. In addition, maybe the C-plane device can be put in the supplementary Information, since it is only part of the structural discussion.

Other remarks/questions:

1. General: in the introduction this material system ZnSe/ZnTe has been motivated for PV applications. I think that for PV, one prefers to have the small bandgap material inside and the large band gap material on the outside.

2. Line 121: ‘ suggested strain-induced switch had hitherto not been observed experimentally to the best of our knowledge’. Although in a different way, a similar transition has been reported for the GaAs system. Inducing a direct-to-pseudodirect bandgap transition in wurtzite GaAs nanowires with uniaxial stress, G. Signorello, E. Lörtscher, P.A. Khomyakov, S. Karg, D.L. Dheeraj, B. Gotsmann, H. Weman & H. Riel , Nature Communications volume

5, Article number: 3655 (2014)

3. Line 185: 'We attribute the formation of different morphologies to the different surface energies of C and A sapphire planes, which modify the contact angle of the catalyst and thus influence the final morphology of the deposited material.' It is not directly clear from the main text that the wires on the different substrates have the same crystal orientation. This can only be seen in the figure.

4. Line 193: the green & blue color coding is poorly visible in figure 2d.

5. Line 211: the authors are asked to indicate the positions of the misfit dislocations in figure 2h, as i) they do not show up at first sight in such a small image, and ii) a reader unexperienced with GPA rotation maps might not be able to recognize these features in such an image.

6. Line 212: can the authors provide direct evidence of the misfit dislocations by displaying atomic resolution images of the corresponding areas?

7. Line 212: Does the phrase 'especially well visible' imply that there are also 'less well visible' misfit dislocations, e.g. in figure 2g?

8. Line 213: To me it is not clear what the 'gradual plane rotation' means and where I could see it in figure 2h.

9. Line 219: Figure 2i is not clear. Are these line traces constructed from figures 2g and 2h? If so, explain or draw lines in these images at the corresponding positions.

10. Line 225: '... n of 0.2% in the samples grown on A-plane sapphire and 0.1% in those grown on...' is this obtained from GPA? What is the accuracy of GPA?

11. Line 414: "gap values per region suggests a band type transition from direct to indirect of ZnTe under compressive strain (Figure SI14) and supports what was predicted theoretically by DFT calculations in the literature.⁸¹' Could this be explained and quantified in the text? It is not clear which bands have shifted with strain, and if the obtained strain levels could fully explain this transition.

Response to referees

We would like to express our sincere appreciations for reviewers' constructive comments concerning our manuscript entitled "**Sub-nanometer mapping of strain-induced band structure variations in planar nanowire core-shell heterostructures**" (Manuscript No.: NCOMMS-22-01047-T).

These comments are all valuable and helpful for improving our manuscript. According to the reviewers' comments, we have added the requested modifications to our manuscript and supplemented extra data to improve our manuscript. In the revised version, changes to our manuscript were all highlighted within the document by using **blue colored text**. Point-by-point responses to reviewers are listed below.

NOTE: *Text in bold and italics* show the original comments addressed by the referees. Our answers have been written in plain black text, while the additions/modifications to the main/supplementary text addressing the reviewers' issues have been added in blue.

Reviewer #1:

The manuscript addresses the structural interplay at the interface between core and shell of planar arrays of core-shell semiconductor nanowires where the strain is not as easily released as in free standing nanowire. The manuscript presents impressive experimental data and careful data analyses and modeling to extract detailed information about the spatial variation in strain at the interface.

The structural rearrangement at the core-shell interface and spatial variation in strain affect the band gap and also causes a local transition from direct to indirect band gap as an effect of compressive strain on the sub-nanometer scale.

The methodology is sound, the work is of high quality and well described.

- We thank Reviewer #1 for the positive support to our manuscript.

I have one question about the data presented in Figure 1c. There is a contrast variation in the shell where thin lines of brighter contrast are seen stretching from the corners of the core to the surface. A comment on this would be welcome.

- The original image shows intensity saturation in some areas due to camera brightness adjustments. We have changed the Fig1c image by another raw image of the same nanowire with other brightness/contrast settings which does not reveal any anomalous section.

The observations are of high significance to the field. I recommend this manuscript to be accepted for publication.

- We thank again Reviewer #1 for the supportive opinion on our manuscript.

Reviewer #2:

Papier: “Sub-nanometer mapping of strain-induced band structure variations in planar nanowire core-shell heterostructures” by Sara Martí-Sánchez et.al., present interesting attempt of mapping of local bandgap in strained laterally grown nanowire. The authors claim to link the experimentally determine spatially resolved bandgap with local strain in thin cross-sections of such heterostructure.

The general remarks:

1. The detail of procedure/method of determination of local bandgap should be shifted in condensed form from supplement to main paper. This part is critical in support of the one of the thesis of the article: “Our analysis reveals a decrease in band gap energy decrease at highly strained core-shell interfacial regions, along with a switch from direct to indirect band gap, as inferred by fitting to their corresponding spectral functions”.

- We agree with the reviewer observation, also pointed by reviewer 3. In response, the following text was added to the body of the main manuscript, in the “Band gap measurements through Valence EELS” section, to summarise the main aspects of the methodology. In the added text, we have also mentioned some of the key points that arose in the present review process. Additionally, these issues have their own specific additions throughout the main and supplementary text. These are individually and specifically discussed in the corresponding reviewed point. Complementarily, we included a figure to graphically support the text describing the methodology.

The new text introduces that the major details can now be found in the main text:

“With this approach, we can subtract the parasitic contributions in the experimental EELS data and perform meaningful fittings to extract physically representative information. The details on the applied methodology are described next, while further details can be found in the Supplementary Information (Section SI7).”

After one paragraph, the following text has been added:

“The ability to spatially identify the pixels corresponding to each of the materials of interest allows the subsequent tailored-data processing assisted by STEM-VEELS simulations of the device. These simulations are based on classical electrodynamics, with the materials described by their frequency-dependent, local dielectric functions. The energy-loss probability is directly separated as the sum of a bulk contribution, which is independent of geometry and proportional to the path length travelled inside each material, and a surface term determined by the interface morphology.⁸¹ We perform both retarded and non-retarded (i.e., assuming an infinite speed of light) calculations and compare them to assess the role of Cherenkov radiation emitted by the fast electrons in their interaction with the bulk

materials. Given the translational invariance of our system along the electron-beam direction, the bulk contribution (Γ_{Bulk}^R) is the same as for an electron moving in an infinitely extended material, while the interface corrections (Γ_{Surf}) are determined by solving Maxwell's equations with the boundary-element method.⁸² We find converged results when the interfaces are parametrized with points separated by 0.3-0.5 nm, which incidentally corresponds to the spatial resolution in our experimental spectral images. Although this precision does not reach atomic resolution, it allows to convolve the atomistic effect of dislocations. Quantum confinement effects are safely ignored because of the large length scale (~100 nm) involved in the nanowire structure. Our simulations assume perfect crystals, although complementary simulations demonstrate the robustness of the method with respect to strain and a bandgap change smaller than ± 1 eV (Supplementary Information, Sections SI7 and SI8).

Based on the information provided by the simulated retarded and non-retarded spectral images, we correct the experimental data to subtract parasitic contributions associated with Cherenkov radiation and excitation of surface modes. By geometrically correlating the simulated device shape with the experimental one, a correction function is pixelwise applied to the experimental spectral image to only retain the experimental loss probability corresponding to the bulk excitations of interest. The correction function, which depends both on the position (pixel) and on the energy (spectrum channel), is the ratio between the non-retarded bulk loss (Γ_{Bulk}^{NR}) and the total retarded loss ($\delta(\mathbf{r}', \omega) = \Gamma_{Bulk}^{NR} / \Gamma_{Total}$, where $\Gamma_{Total} = \Gamma_{Bulk}^R + \Gamma_{Surf}$). Each of the terms involved in the correction factor depends linearly on the lamella thickness, making the process robust against thickness modulations or inhomogeneities. In our case, the lamella thickness was estimated to be 45 nm thick and homogenous. The entire workflow of data cleaning and processing assisted by the simulations is described in Figure 4. The detailed algorithmics behind these corrections are discussed in the Supplementary Information (Sections SI7 and SI8).

The so-cleaned experimental data (free of interfering Cherenkov and surface-related signals) are ready to perform a meaningful analysis. We proceed by implementing adaptive and material-dependent curve fittings in a pixelwise fashion to determine the band-gap structure (direct or indirect) that better represents the nature of the observed spectra.⁶⁷ Our method is not limited to solely determining the band gap value; ...".

As said, we have also added new figure, now being Figure 4, to graphically support the text describing the methodology, see below:

Figure 4. Workflow of the developed method for correcting the measured low-loss Electron Energy-Loss Spectroscopy (EELS) data affected by Cherenkov radiation and surface modes. Classical electrodynamics simulations are used to produce a spectral image of both the non-retarded and retarded loss probabilities, which are by construction decomposed into bulk and surface contributions. Next, a correction function that compares the simulated non-retarded and retarded losses is used to map the effect of parasitic radiation and to clean the experimental spectral image. Finally, the corrected experimental signal is used to fit either a direct or indirect band-gap structure depending on the material and spatial position, and to extract a band-gap value from such fit.

2. *All measurements are performed on thin FIB cross-section. However, the impact of lamella thickness (which can be mapped by EELS), on both, the local strain and bandgap, are not taken into account or discussed in analysis/EELS model.*

The reviewer raises an interesting discussion point regarding the effects of the lamella thickness. The effect of thinning a lamella should induce a relaxation in the direction through the lamella direction allowing the other two components to relax as well and show an apparent reduction of the elastic strain. In the studied case, the high mismatch between the core-shell structures makes most of the relaxation to happen through misfit dislocations. Therefore, the remaining elastic component that could be relaxed by the previous phenomenon is small and would make the corrections of considering this effect negligible. Indeed, as we claim, the elastic component is released with the relative rotation

of the shell planes, which would be difficult to reverse given the equilibrium found between the dislocations and elastic strain (as proved with the models). Furthermore, in this case, the lamella thinning would not be a strong enough stress effect to relax the dislocations by, for instance, dislocation drift, as their location is deterministic despite the thickness, as proved by the provided atomistic and geometrical models. The expected low influence of thickness in this case to strain, and as a result to the computed bandgap, takes us to consider this source of uncertainty to levels even below the already provided uncertainty. For instance, in this case, the error propagation associated to the model fitting would be the largest uncertainty source.

In any case, any effect would be uniformly applied throughout the full spectrum image as the thickness mapping revealed a uniform thickness distribution. This analysis was done by applying both the relative and absolute thickness mapping in Digital Micrograph for the used spectrum image. The relative thickness map showed that, within regions, the thickness computed is uniform (Figure SI16). Therefore, the variations in the colour scale between materials, observed in Figure SI16 are attributed to differences in the inelastic mean free path, which is material dependent, and not thickness. Moreover, the absolute thickness computation, performed for the main materials ZnSe and ZnTe, revealed comparable average thicknesses, $44\pm 3\text{nm}$ and $46\pm 1\text{nm}$, respectively.

Figure SI16. Relative thickness map obtained by using the Digital Micrograph algorithm, in units of the inelastic mean free paths.

Furthermore, the EELS methodology is robust to thickness variations, as proved below:

Practically, the theoretical simulations affect the experimental spectrum image by applying to every pixel and energy channel a correction factor. This value ranges from 0 to 1 and is computed in the following way (details of the equations in the supporting information):

$$\text{Correction ratio} = \delta(\mathbf{r}', \omega) = \frac{\Gamma_{Bulk}^{NR}}{\Gamma_{Total}} ; \text{ where } \Gamma_{Total} = \Gamma_{Bulk}^R + \Gamma_{Surf}$$

Where:

$$\Gamma_{bulk}^R(\omega) = \frac{Le^2}{\pi\hbar v^2} \text{Im} \left\{ \left(\frac{v^2}{c^2} - \frac{1}{\epsilon} \right) \ln \left(\frac{q_c^2 - k^2\epsilon}{\left(\frac{\omega}{v}\right)^2 - k^2\epsilon} \right) \right\}$$

$$\Gamma_{bulk}^{NR}(\omega) = \frac{2e^2L}{\pi\hbar v^2} \text{Im} \left\{ -\frac{1}{\epsilon} \right\} \ln \left(\frac{q_c v}{\omega} \right)$$

and Γ_{Surf} comes from the simulations based on classical electrodynamics, that returns a Γ_{Surf}' with units of $\text{eV}^{-1}\text{nm}^{-1}$. To translate the loss probability per thickness unit, we just multiply by the thickness:

$$\Gamma_{Surf} = L\Gamma_{Surf}'$$

As a result, given that each of the terms contributing to the correction ratio present the same linear dependence with their corresponding loss probability per thickness unit ($\Gamma_{Bulk}^{NR'}$, Γ_{Bulk}^R , Γ_{Surf}'), the correction ratio becomes independent to the thickness:

$$\delta(\mathbf{r}', \omega) = \frac{\Gamma_{Bulk}^{NR}}{\Gamma_{Total}} = \frac{\Gamma_{Bulk}^{NR}}{\Gamma_{Bulk}^R + \Gamma_{Surf}} = \frac{L\Gamma_{Bulk}^{NR'}}{L\Gamma_{Bulk}^R + L\Gamma_{Surf}'} = \frac{\Gamma_{Bulk}^{NR'}}{\Gamma_{Bulk}^R + \Gamma_{Surf}'}$$

By knowing this dependence, the calculations were performed using an arbitrary thickness representative of a typical lamella as can be 60nm (the exact number used was 57.5nm). Therefore, the results and plotted curves directly considering the loss probabilities per thickness unit would be equally meaningful.

With the added explanation, provided in point 1, the following text addresses this issue in the main manuscript text:

The correction function depends on the position (pixel) and on the energy (energy channel) and is the ratio between the non-retarded bulk loss (Γ_{Bulk}^{NR}) and the total retarded loss ($\delta(\mathbf{r}', \omega) = \Gamma_{Bulk}^{NR} / \Gamma_{Total}$, where $\Gamma_{Total} = \Gamma_{Bulk}^R + \Gamma_{Surf}$). Every term involved in the correction factor depends linearly on the lamella thickness, making the process independent of thickness modulations or inhomogeneities. In any case, the lamella thickness was estimated to be 45 nm and homogenous throughout. The whole workflow of the data cleaning and processing based on the simulations is described in Figure 4.

To provide extra insight on this issue, we added the following text in the supporting information, in section SI8. Methodology for band gap mapping.

“This ratio is always a value between 0 and 1, and it represents the amount of signal or loss probability that comes intrinsically from the material at a given energy, ω , and position of the simulated spectrum-image, r' , (in other words, signal free of all kinds of additional contributions), compared to the total signal or loss probability that also includes Cherenkov radiation and waveguide modes arising from the interfaces. Note that the correction ratio involves loss probabilities that depend linearly on the thickness of the considered device. More precisely, the surface term obtained in the simulation is a loss probability per unit length along the thickness of the sample. Upon multiplication by the lamella thickness, each of the three terms in the ratio has the same linear dependence on thickness, thus making the ratio independent of this variable and robust against sample-to-sample thickness variations. In the present instance, the thickness was determined for ZnSe and ZnTe to have a uniform distribution around 45 nm. This is corroborated by the smooth thickness regions observed in Figure SI16, where variations in colour scale between materials are attributed to differences in the material-dependent inelastic mean free path, and not thickness changes. Moreover, the absolute thickness estimate, performed for the main materials ZnSe and ZnTe, revealed comparable average thicknesses of 44 ± 3 nm and 46 ± 1 nm, respectively.”

3. What is the impact of interface misfit dislocations that can act as low bandgap centers? What about influence of surface states related to FIB radiation damage/amorphisation of lamella surfaces?

- The reviewer also raises an interesting discussion with this observation. The individual effect of the dislocations in the ZnSe-ZnTe interface cannot be checked by the limitations in the spatial resolution we have used in our experimental setup. Although we obtained a sub-nanometric pixel size, it did not reach atomic resolution. Therefore, an atomistic effect such as misfit dislocations will not be observed in our map. We might see their spatially pooled effect together with the remaining elastic strain of the interface, but we are unable and not aiming to separate the contributions given the resolution limitation. As a result, we describe the result of this combined effect. In fact, we can confirm that the observed band modulation source is twofold, as the interfaces described in the following theoretical paper are free of dislocations and caused by elastic strain (R. Peköz and J.-Y. Raty *Phys. Rev. B* **84**, 165444), being the remaining source, the dislocations acting as low bandgap centres.

To adapt the model to this issue, atomistic simulations going to *ab initio* levels of theory would be necessary to take into account the effect these dislocations can have on the computed correction factor (as in the following reference: A. Gutiérrez-Sosa *et al. Phys. Rev. B* **66**, 035302). This superior level

of theory would only be beneficial if the resolution of our map could take this atomistic extension into account, as our resolution would average and dilute the computed precision. As a result, the boundary element nature of our simulations constitutes a first approximation that fits well the requirements of our experimental setup.

We have added the following text to pinpoint this idea in the manuscript, in the added paragraphs on the methodology (main manuscript):

“Given the translational invariance of our system, the bulk contribution (Γ_{Bulk}^R) matches an electron in an infinitely extended crystal, and the interface corrections (Γ_{Surf}) can be accounted for by solving Maxwell’s equations with the boundary-element method. The spatial resolution of the simulations is of 0.3-0.5 nm between parametrization points, which fits well the spatial resolution of an experimental spectral image that does not reach atomic resolution and can only convolve the atomistic effect of dislocations. Quantum confinement effects are not considered given the large scale of the nanowire, around 100 nm.”

We have also added new text in Section SI7:

“Given the large scale of the features in the nanowire (around 100 nm) we do not expect quantum confinement effects, which would modify the dielectric response of the involved materials, to play a significant role. Therefore, the bulk dielectric functions used as reference for the corrections should be suitable for our study. Incidentally, the BEM discretization used to obtain converged simulations in commensurates with the spatial resolution of our measured spectral images. Therefore, atomistic effects such as dislocations are not considered within the simulated model used for correction of the experimental data. In fact, the effect of dislocations as low-band-gap centres adds up to the elastic strain to average the response that we observe in the band-gap maps. Atomistic effects could be eventually incorporated, for instance, via *ab initio* simulations similar to those employed in the context of atomic-resolution EELS spectral imaging.¹¹”

Moreover, the section of supplementary information “Strain considerations”, under Section SI7, quantitatively proves the validity of the correction method under a large variation of the dielectric functions of both materials because of strain. It proves the correction would still be valid under a big change of bandgap of up to 5eV for both ZnSe and ZnTe. The nature of the corrections (i.e. keeping a percentage of the original curve) makes a hypothetical decrease, smaller than 1eV, in ZnTe bandgap manageable by keeping the nature of the band type curve (1eV is the spectral range that trades off the bandgap model fitting, avoiding overfitting and dealing well with noise). This range of validity is much larger than the scale of the change we observe (average around 40meV). Moreover, the hypothetical decrease in the ZnTe bandgap would have no effect to the dielectric response of ZnSe as this material is transparent in that spectral range.

We have also added the following text to pinpoint this idea in the manuscript:

In the main text, in the added paragraphs on the methodology:

“Quantum confinement effects are not considered given the large scale of the nanowire, around 100 nm. The simulations consider perfect crystals, although complementary simulations have proved the robustness of the method with respect to strain and a bandgap change smaller than ± 1 eV (supplementary information, Sections SI7 and SI8).”

And also, at the end of Section SI8:

“However, the fitting process still captures the essence of the experimental signal and allows to properly adapt the direct curve confirming this trend. Finally, Figure SI18j belongs to the sapphire region. Here we see how the surface modes observed in Figure SI18d are completely removed and the curve indicating the real states becomes clear together with its indirect band structure fitting. Regarding curve fitting and the tolerance of the method to band-gap changes, the pixelwise evaluation of the fitted spectral ranges, revealed a trade-off between overfitting and noise handling at 1eV. Consequently, we expect that any band-gap shift of ± 1 eV with respect to tabulated values to be properly captured by the model. Indeed, this range would be enough to resolve most of the phenomena that can induce band shifts. For larger band-gap shifts, it would be difficult to ensure the validity of the method that keeps the reference value as the one tabulated. In order to generalise this method and adapt it to larger variations, simulations should also consider band modulation by strain (e.g., from *ab initio* calculations).”

Regarding the damage induced by the FIB during the lamella preparation and its influence to the EELS map, the gentle conditions in which it was prepared should prevent any surface state to be significant in neither the imaging nor the spectroscopy. The lamellae preparation was done in a FIB FEI Helios 650, equipped with a gallium ions beam, and operated at progressively decaying voltages (from 30kV to 1kV) to ensure the minimum surface damage and ion implantation. The literature suggests that for lamella preparation at voltages below 2kV, the generated amorphous layer will be at the few nm range. More precisely, Mayer, J. *et al.* observed an (sample dependant) amorphous layer thickness of 0.5-1.5nm when finishing the lamellae thinning with 2kV [1]. Therefore, the operation at 1-2kV used in our experimental setup would imply a thickness below the nm range. In the reported conditions, the influence of the sample modification in the final image or spectroscopic signal should be negligible according to the literature. In addition, a protective Pt layer was used in the lamellae fabrication to further prevent ion implantation and sample damage.

The following text has been included in the main text, in the methodology section, to inform about the capabilities of the FIB and how it can prevent the discussed issue:

“(S)TEM Imaging and Spectroscopy – The transversal and longitudinal cross sections were prepared by a Focused Ion Beam (FIB) FEI Helios 650 and HAADF imaging was performed in a FEI TITAN Low-Base operated at 300 kV. Lamellae were prepared using Ga-ion FIB operated at voltages ranging between 30kV in the first stages to 1kV in the last steps. This procedure reduced FIB-induced damage and Ga implantation to a minimum, also resulting in an amorphized layer thickness below 1 nm, which had a negligible impact on the imaging and spectroscopy analyses. EELS elemental maps were obtained in a FEI TECNAI F20 TEM microscope operated at 200 kV.”

Supporting literature:

- 1- Mayer, J. *et al.* MRS Bulletin 32, 400–407 (2007). <https://doi.org/10.1557/mrs2007.63>
- 2- McCaffrey, J.P. *et al.* Ultramicroscopy 87 (2001) 97–104
- 3- Schaffer, M *et al.* Ultramicroscopy 114 (2012) 62–71
- 4- Ishitani, T *et al.* Journal of Electron Microscopy, 53, 5 (2004) 443–449, <https://doi.org/10.1093/jmicro/dfh078>

4. Whether the use of the term "crystallinity, which is gaining more and more popularity is appropriate here for defective but monocrystalline heterostructure? Originally, this term described the ratio of the crystalline phase to the amorphous phase in materials such as polymers determined by X-ray diffraction. Authors have details information about structure from TEM. So more precision descriptors can be used.

- We employed twice the term crystallinity in the introduction section referring to monocrystallinity. We have changed the term by “high crystal quality” or “monocrystalline material heterostructure” to be more precise with the description.

“The nanometer-scale lateral dimensions and resulting quasi one-dimensional morphology of NWs facilitate the release of the inherent epitaxial strain, preserving high ~~crystallinity~~ crystal quality even when interfacing highly mismatched materials.²⁰ In NWs, plastic and elastic relaxation mechanisms can combine to release strain during axial growth while maintaining high ~~crystallinity~~ a monocrystalline material heterostructure^{20,21}, thus introducing (...)”

5. At Figure 1a the indication of crystallographic direction should, by convention be [-101], not [-202]. What material is present in the black area between FIB platinum and NW in fig 1c?

- We have modified the direction in Figure 1a following Referee's suggestion. Both directions [-101] and [-202] are equivalent, but we agree with the referee that by convention it is better to use [-101].

See the modification added in **Figure 1a**.

The material present in the black area between FIB platinum and NW is amorphous SiO_2 . It is deposited with two main purposes: NW protection during FIB preparation and improving image contrast on top of the heterostructure.

6. Authors claim that sapphire orientations determine orientation of laterally grown NW. Nevertheless, high-resolution images of Sapphire/NW interface are not given. The sapphire substrate looks like masked in figure 1c, 2a,b. As well, Bragg peaks from sapphire are not present on FFT fig 2 c,d allowing confirmation of epitaxial relations on NW and substrate. The low magnification image of substrate/NW can only be seen in figure S113. Authors should include high-resolution images of sapphire/NW boundaries to prove that this interface is semi coherent and force orientation of NW crystal lattice.

- The NW guided growth mechanism described in the manuscript had already been reported by us in more detail in a previous manuscript, cited as ref. [46] in the present paper (ACS Nano, 11, 6155 (2017)). In the present manuscript we were only describing the growth mechanism briefly in order to illustrate the problem of interest, which is the in-depth sub-nanometer study of the strain-induced band structure variations in our core-shell nanowires. Certainly, as the referee says, we claim that sapphire orientations guide the NW growth direction on the substrate, while NW orientations are due to the

VLS growth, typically creating growth fronts on the $\{111\}$ planes and a $\{112\}$ overall crystal growth direction (see further details in ref. [46]). In the present manuscript, we wanted to illustrate this fact in Figure 1e, where we observe a $\{111\}$ faceting at the edge of the longitudinal cross section, while the direction in which the NW elongates is $\{112\}$. This growth direction is observed for both substrate orientations examined, as seen from the FFT of the cross-sections. The substrate guides the direction of the growth on substrate planes, as seen in Figure S1, creating several parallel NWs in the main sapphire crystallographic directions.

Regarding the atomic resolution images of the substrate, we must say that in the present manuscript we did not show any of them. The reason why is because we wanted to emphasize the atomic resolution of the NW core and shell by using high quality aberration corrected HAADF STEM images. As it is well known, HAADF STEM images show an intensity dependent on Z^{-2} . In our case, as one of the materials is composed of light elements (such as the sapphire substrate Al_2O_3), if we wanted to obtain a clear image of the light substrate atomic structure, we had to saturate the part of the image corresponding to the core and shell (composed of much heavier elements: ZnTe/ZnSe). In order to avoid it, we obtained the HAADF STEM images focussing on the improvement of the core and shell atomic resolution contrast. However, in order to demonstrate the epitaxial substrate/NW relationship asked by Referee #2, we show the following HRTEM images, where the contrast is not Z^{-2} dependent, and their corresponding FFTs. In this case, all, the substrate (sapphire), core (ZnSe) and shell (ZnTe), lattice spots clearly appear in the FFTs.

Figure 2 from ref. [46]. Guided growth of epitaxial core-shell nanowires on C-plane and A-plane sapphire. (A,B) For each substrate, respectively: (a) schematic illustration of the plane and directional growth; (b) SEM image of the guided nanowires; (c) HRTEM cross-section image, marked with the

crystal planes and direction of the nanowire ZnSe core (yellow), ZnTe shell (red), and the sapphire substrate (blue); (d) selected area fast Fourier transform of the cross-section image.

7. The main finding of the paper is related to interface region between ZnTe shell and ZnSe core. Geometric models and GPA analysis suggest the presence of misfit dislocations at these interfaces. What is the nature of these dislocations (Lomer, 60 degree etc..) Authors present the high clarity filtered, atomic resolution STEM images of “atoms arrangement “in perfect part of NW (fig 1d,e). The example of high resolution image with interface area containing misfit dislocation allowing a determination of projected components of Burger’s vector of these dislocations should be given.

We agree that we did not add any supporting material apart of GPA showing a direct visualization of dislocations on the interface. For that reason, and with the aim to be easier to follow by readers non-experienced with GPA, and following Referee #2 suggestions, we have added a new section in the SI (SI6. Details on the Core-shell misfit dislocations) with the addition of atomic resolution direct evidence of this dislocations. Our samples present two different types of misfit dislocations visible on our projected visualization axis:

- 1) On the lateral sides of the NW core-shell interface, misfit dislocations consist of the addition of a full (111) plane in the core with respect to the shell.
- 2) On the top side of the NW core-shell interface, misfit dislocations consist of the addition of half (10-1) plane in the core with respect to the shell.

Both of these defects are pure edge dislocations. In Figures SI11 and SI12 we pointed the dislocation positions in GPA maps to help readers interpret their position in rotational maps and at the same time we have included their position and direction in the atomic resolution HAADF STEM images zooming them in detail. For clarity, we have applied a frequency filter to the HAADF STEM images.

Figure SII1. Details of dislocation formation on (111) and (20-2) planes on a C-plane NW core-shell lateral and top interfaces, respectively. HAADF STEM micrographs zooming the core-shell interface, GPA rotation maps of (111) and (20-2) planes with arrows indicating dislocation positions and zoom in of FFT plane filtered micrograph in the region marked in the HAADF STEM images indicating the dislocation configuration.

Figure SII2. Details of dislocation formation on (111) and (20-2) planes on a A-plane NW core-shell lateral and top interfaces, respectively. HAADF STEM micrographs zooming the core-shell interface, GPA rotation maps of (111) and (20-2) planes with arrows pointing interface dislocations and zoom in of FFT plane filtered micrograph in the region marked in the HAADF STEM images indicating the dislocation configurations.

8. The discussion about, the impact of the sapphire on strain distribution in NW determined by GPA should be added. GPA measurements give relative values of lattice distortion to the reference area taken in the center of ZnSe core. Are the NWs fully relaxed in relation to sapphire substrate?

As said before, the high Z^2 average difference between ZnSe and ZnTe in comparison to sapphire (Al_2O_3) saturates the contrast and makes sapphire invisible in HAADF STEM images, the technique of choice for our GPA analyses as it provides low environmental noise. Therefore, we could not simultaneously image both materials in HAADF STEM. However, by using our HRTEM images, we could measure the lattice constants of the core in comparison to the substrate in several NWs, and we could find that the ZnSe core lattice parameters matched with its relaxed value, so we could take this value as a reference in our GPA maps and measure the distortions caused in the shell with respect to the core reference.

The following paragraph has been added in the main manuscript text (page 12):

“The high Z^2 average difference between ZnSe and ZnTe in comparison to sapphire (Al_2O_3) saturates the contrast and makes sapphire invisible in HAADF STEM images, the technique of choice for our GPA analyses as it provides low environmental noise. Therefore, we could not simultaneously image both materials in HAADF STEM. However, by using HRTEM images, we measured the lattice constants of the core in comparison to the substrate in several NWs, and we found that the ZnSe core lattice parameters matched with its relaxed value, so we could take this value as a reference in our GPA maps and measure the distortions caused in the shell with respect to the core reference.”

9. It seems that some references in text to the figures in supplement are not valid. For example, there are two figures indicted as SI3 in supplement.

- Thank you for noticing it, we have renumbered the Figures in the Supporting information and verified the rest of the document.

10. The sentence “Examining this area in the GPA calculations and MAADF-STEM contrast images suggests that a high degree elastic strain remains there.” is used twice in the same paragraph.

Thanks for noticing it. The first sentence has been removed, leading to the following paragraph:

“Thus, we tried the fitting for the VEELS corresponding pixels in this region with an indirect band gap function. ~~Examining this area in the GPA calculations and MAADF-STEM contrast images suggests that a high degree elastic strain remains there.~~ The indirect fitting offers a better performance in terms of meaningful resulting physical information and in goodness of fit, significantly surpassing the direct fitting with a confidence level higher than 99.9995% (see “Significance tests” in the Supplementary Information). Examining this area in the GPA calculations and MAADF-STEM contrast images suggests that a high degree elastic strain remains there. The details on this evaluation can be found in Section SI8.”

(See next page for the answers to Referee #3)

Reviewer #3:

This is a high-level paper, which is well written, presenting a timely topic. I would support publication of this work, but the balance and the accessibility of the paper may be optimized: a lot of text has been spent on Figures 1 and 2 (up to page 12), while these present a rather straightforward characterization of the material system. Maybe this first part can be shortened. I find Figures 3 and 4 the most interesting, but also difficult to follow. A background on TEM is needed, and there are many references to the supplementary information. Maybe more explanations can be included in the main text in this second part. In addition, maybe the C-plane device can be put in the supplementary Information, since it is only part of the structural discussion.

We thank Reviewer #3 for the support given to our work. Following Reviewer suggestions, we have moved some part of the manuscript initial sections dealing with the description of Figures 1 and 2 to the supporting information, and we have strengthened the second section part, that we agree is the key part of our manuscript. E.g.: We have reduced 1 page in the description of Figures 1 and 2 and moved it to a new section in the Supporting Information (**SI5. Additional details on the strain relaxation mechanisms and shell rotation**).

We agree that a deeper explanation of the second section of the paper must be provided to support the findings and latter figures, as also pointed out by Reviewer #2. In this way, we have included the following text in the body of the main text, in the “**Band gap measurements through Valence EELS**” section, to summarise the main aspects of the methodology and address the raised issues of the present revision process. In addition, we have included a new Figure complementing the explanation of this methodology.

The new text introduces that the major details can now be found in the main text:

“With this approach, we can subtract the parasitic contributions in the experimental EELS data and perform meaningful fittings to extract physically representative information. The details on the applied methodology are described next, while further details can be found in the Supplementary Information (Section SI7).”

After one paragraph, the following text has been added:

“The ability to spatially identify the pixels corresponding to each of the materials of interest allows the subsequent tailored-data processing assisted by STEM-VEELS simulations of the device. These simulations are based on classical electrodynamics, with the materials described by their frequency-dependent, local dielectric functions. The energy-loss probability is directly separated as the sum of a bulk contribution, which is independent of geometry and proportional to the path length travelled inside each material, and a surface term determined by the interface morphology.⁸¹ We perform both

retarded and non-retarded (i.e., assuming an infinite speed of light) calculations and compare them to assess the role of Cherenkov radiation emitted by the fast electrons in their interaction with the bulk materials. Given the translational invariance of our system along the electron-beam direction, the bulk contribution (Γ_{Bulk}^R) is the same as for an electron moving in an infinitely extended material, while the interface corrections (Γ_{Surf}) are determined by solving Maxwell's equations with the boundary-element method.⁸² We find converged results when the interfaces are parametrized with points separated by 0.3-0.5 nm, which incidentally corresponds to the spatial resolution in our experimental spectral images. Although this precision does not reach atomic resolution, it allows to convolve the atomistic effect of dislocations. Quantum confinement effects are safely ignored because of the large length scale (~100 nm) involved in the nanowire structure. Our simulations assume perfect crystals, although complementary simulations demonstrate the robustness of the method with respect to strain and a bandgap change smaller than ± 1 eV (Supplementary Information, Sections SI7 and SI8).

Based on the information provided by the simulated retarded and non-retarded spectral images, we correct the experimental data to subtract parasitic contributions associated with Cherenkov radiation and excitation of surface modes. By geometrically correlating the simulated device shape with the experimental one, a correction function is pixelwise applied to the experimental spectral image to only retain the experimental loss probability corresponding to the bulk excitations of interest. The correction function, which depends both on the position (pixel) and on the energy (spectrum channel), is the ratio between the non-retarded bulk loss (Γ_{Bulk}^{NR}) and the total retarded loss ($\delta(\mathbf{r}', \omega) = \Gamma_{Bulk}^{NR} / \Gamma_{Total}$, where $\Gamma_{Total} = \Gamma_{Bulk}^R + \Gamma_{Surf}$). Each of the terms involved in the correction factor depends linearly on the lamella thickness, making the process robust against thickness modulations or inhomogeneities. In our case, the lamella thickness was estimated to be 45 nm thick and homogenous. The entire workflow of data cleaning and processing assisted by the simulations is described in Figure 4. The detailed algorithmics behind these corrections are discussed in the Supplementary Information (Sections SI7 and SI8).

The so-cleaned experimental data (free of interfering Cherenkov and surface-related signals) are ready to perform a meaningful analysis. We proceed by implementing adaptive and material-dependent curve fittings in a pixelwise fashion to determine the band-gap structure (direct or indirect) that better represents the nature of the observed spectra.⁶⁷ Our method is not limited to solely determining the band gap value; ...”.

As said, we have also added new figure, now being Figure 4, to graphically support the text describing the methodology, see below:

Figure 4. Workflow of the developed method for correcting the measured low-loss Electron Energy-Loss Spectroscopy (EELS) data affected by Cherenkov radiation and surface modes. Classical electrodynamics simulations are used to produce a spectral image of both the non-retarded and retarded loss probabilities, which are by construction decomposed into bulk and surface contributions. Next, a correction function that compares the simulated non-retarded and retarded losses is used to map the effect of parasitic radiation and to clean the experimental spectral image. Finally, the corrected experimental signal is used to fit either a direct or indirect band-gap structure depending on the material and spatial position, and to extract a band-gap value from such fit.

Other remarks/questions:

1. General: in the introduction this material system ZnSe/ZnTe has been motivated for PV applications. I think that for PV, one prefers to have the small bandgap material inside and the large band gap material on the outside.

The material combination provides a radial pn junction heterostructure, having then a larger area of interaction between both materials. The reason for such ZnSe core – ZnTe shell combination arises from growth conditions for the materials. ZnTe can be deposited on ZnSe at a lower temperature, making this combination feasible given the growth conditions of both materials.

The following sentence has been added to the main manuscript Introduction:

“The reason for such core-shell combination arises from the fact that ZnTe can be deposited on ZnSe at a lower temperature, making this combination feasible given the growth conditions of both materials.⁴⁶”

2. Line 121: ‘suggested strain-induced switch had hitherto not been observed experimentally to the best of our knowledge’. Although in a different way, a similar transition has been reported for the GaAs system. Inducing a direct-to-pseudodirect bandgap transition in wurtzite GaAs nanowires with uniaxial stress, G. Signorello, E. Lörtscher, P.A. Khomyakov, S. Karg, D.L. Dheeraj, B. Gotsmann, H. Weman & H. Riel, Nature Communications volume 5, Article number: 3655 (2014)

The suggested publication has been checked and indeed presents comparable results despite a completely different experimental approach. The article results are based on photoluminescence spectroscopy for accessing the band structure, and on Raman spectroscopy to measure the strain. Despite not being precise enough, with the original statement we wanted to stress that this phenomenon had not been observed with a sub-nanometer resolution precision in a confined region of a heterostructure to the best of our knowledge. As we honestly were not aware of the publication referred by the reviewer, which supports the results presented in our paper by acting as a precedent, we have modified the statement to account for this example and reference it accordingly. The new version of the text is the following:

In the manuscript introduction:

“The obtained results highlight the nanoscale electronic band gap modulation due to the high strain at the core and shell interface region, including a switch from direct to indirect band gap in the ZnTe interface region. ~~Such a suggested strain-induced switch had hitherto not been observed experimentally to the best of our knowledge.~~ A similar strain-induced switch was reported in wurtzite GaAs,⁵² although the present study is to the best of our knowledge the first time that such strain-induced effect is observed with sub-nanometer resolution precision in a confined region of a heterostructure. With the present study, we show the importance of detailed atomic-scale crystal evaluation for controlling and tuning optoelectronic properties in horizontal 1-D arrays, which is of great relevance in the fields related to electronics, photonics, quantum applications and photovoltaics among others.”

3. Line 185: ‘We attribute the formation of different morphologies to the different surface energies of C and A sapphire planes, which modify the contact angle of the catalyst and thus influence the

final morphology of the deposited material.’ It is not directly clear from the main text that the wires on the different substrates have the same crystal orientation. This can only be seen in the figure.

We thank this appreciation as we noted we did not mention it specifically in the text. For that reason, we added a brief comment about it when describing the crystallographic system after Figure 1.

“We determine that the overall growth direction of the NW is a $\langle 112 \rangle$ direction (for both A and C-plane substrates)...”

4. Line 193: the green & blue color coding is poorly visible in figure 2d.

We have modified the color labelling of both **Figures 2c** and **d** to create a more visible contrast.

5. Line 211: the authors are asked to indicate the positions of the misfit dislocations in figure 2h, as i) they do not show up at first sight in such a small image, and ii) a reader unexperienced with GPA rotation maps might not be able to recognize these features in such an image.

Following Reviewer suggestion, we have indicated the position of visible misfit dislocations in **Figure 2h** by using black arrows. In addition, we have added a new section to the supporting information with detailed atomic resolution HAADF STEM detailed images on the regions where the misfit dislocations are placed, together with their corresponding GPA rotation maps with indications on dislocation positions and a description of the defect origin and configuration. This new section also provides easier dislocation recognition for readers unexperienced with GPA.

New panels for **Figure 2**.

New SI Figures SI11 and SI12 as described in the answer to **query 7 from Reviewer #2**.

Figure SI11. Details of dislocation formation on (111) and (20-2) planes on a C-plane NW core-shell lateral and top interfaces, respectively. HAADF STEM micrographs zooming the core-shell interface, GPA rotation maps of (111) and (20-2) planes with arrows indicating dislocation positions and zoom in of FFT plane filtered micrograph in the region marked in the HAADF STEM images indicating the dislocation configuration.

Figure SI12. Details of dislocation formation on (111) and (20-2) planes on an A-plane NW core-shell lateral and top interfaces, respectively. HAADF STEM micrographs zooming the core-shell interface, GPA rotation maps of (111) and (20-2) planes with arrows pointing interface dislocations and zoom in of FFT plane filtered micrograph in the region marked in the HAADF STEM images indicating the dislocation configurations.

6. Line 212: can the authors provide direct evidence of the misfit dislocations by displaying atomic resolution images of the corresponding areas?

As already answered in the previous query, a new section has been added to the supporting information with atomic resolution HAADF STEM images of the core-shell NW interfaces. The right panels on the above Figures SI11 and SI12 show direct evidence of the misfit dislocations presence.

7. Line 212: Does the phrase ‘especially well visible’ imply that there are also ‘less well visible’ misfit dislocations, e.g. in figure 2g?

The presence of dislocations in **Figure 2g** cannot be clearly appreciated due to lack of resolution in the image. We must consider that **Figure 2g** image was obtained at a lower magnification to be able to visualize the whole NW width, as the NW core was much wider than in the case of the A-plane example, see the scale bars in **Figures 2a** and **2b** for comparison. To clarify this point and pointing out that the presence of plastic strain relaxation from dislocation formation is present regardless of the core morphology, we have added a new section in SI with details on the interface and easily visible dislocations, as mentioned in the previous questions. In this case, we have added new atomic resolution HAADF STEM images with higher magnification as detailed below.

Figure SI11. Details of dislocation formation on (111) and (20-2) planes on a C-plane NW core-shell lateral and top interfaces, respectively. HAADF STEM micrographs zooming the core-shell interface, GPA rotation maps of (111) and (20-2) planes with arrows pointing interface dislocations and zoom in of FFT plane filtered micrograph in the region marked in the HAADF STEM images indicating the dislocation configurations.

8. Line 213: To me it is not clear what the ‘gradual plane rotation’ means and where I could see it in figure 2h.

Gradual plane rotation referred to the fact that the absolute rotation the A-plane NW shell (compared to the nanowire core) smoothly increases when moving from the lateral side to the top and presents its maximum close to the NW center, where this rotation shifts to the opposite direction. Although the variation is smooth, it can be clearly observed in the color scale map and the obtained profile. Nevertheless, in order to avoid the confusion this sentence might generate away from the point of the measurement, we decided to remove the term “gradual” from the sentence.

9. Line 219: Figure 2i is not clear. Are these line traces constructed from figures 2g and 2h? If so, explain or draw lines in these images at the corresponding positions.

Red and blue arrows have been added in **Figures 2g** and **2h**, respectively, to indicate the position and direction of the rotation profiles. We have also modified the profile colour scale palette to improve the visibility of the arrows and labels on top of the rotation maps.

At the same time, we have added a line referring to the arrow labelling on the figure caption to make this clear.

“... **Black arrows in h** indicate dislocation positions. **i.** (111) plane rotation profiles across the center of the NW shell (following the directions indicated by the red and blue arrows in **g** and **h**, respectively) for relevant core morphologies in both C-plane and A-plane substrates.”

10. Line 225:'. n of 0.2% in the samples grown on A-plane sapphire and 0.1% in those grown on.'
is this obtained from GPA? What is the accuracy of GPA?

The accuracy of GPA depends on the image quality/resolution and the presence of noise which can introduce some virtual fluctuations in the measured plane spacing in different positions. In the particular case of the measurements we are showing, we calculate the lattice fluctuations by calculating the mean of the GPA-obtained values over regions, trying to avoid noisy areas. The standard deviation of these values was found to be below 0.9 %. In any case, the values we measure as a mean are compatible with the lattice constant of relaxed ZnTe under the accuracy of the technique. We have indicated that in the text to avoid any confusion.

11. Line 414: "gap values per region suggests a band type transition from direct to indirect of ZnTe under compressive strain (now Figure 5f and further details in Section SI8) and supports what was predicted theoretically by DFT calculations in the literature.⁸¹' Could this be explained and quantified in the text? It is not clear which bands have shifted with strain, and if the obtained strain levels could fully explain this transition.

The reviewer opens an interesting discussion that finds the limits of EELS as an experimental technique. The low-loss EELS regime shows a single bandgap as the sum of the interband energy transfers of the acquired momentum transfer (i.e. used aperture), which just by EELS cannot be determined with a single EELS geometry as used here. One could attempt a full momentum-resolved EELS experiment (trying to keep spatial resolution in the nm range). But clearly this is beyond this work for now. The use of theory or complementary experimental techniques, or both (as in the article discussed in **Reviewer #3's Query 2**, see above), would be needed to fully address this question. To go through this limitation, we support our statements by the theoretical study we cite in Ref. [84]: Peköz R. & Raty J. Y. Band structure modulation of ZnSe/ZnTe nanowires under strain. Physical Review B. 84, 165444 (2011). The study revealed that ZnTe under compressive strain can switch from direct to indirect bandgap due to the conduction band minimum in the Brillouin zone Z point reducing its energy (getting closer to the Fermi level). This makes that the smallest interband energy goes from the Γ points in both valence and conduction bands (direct), to Γ in valence to Z in conduction (indirect). The theoretical study forces a change in the lattice parameter, while in our case the strain is caused by the heterojunction. Furthermore, this paper does not consider the presence of misfit dislocations in any of the studied cases. The provided geometrical models revealed that the shell bending is originated to minimise the number of dislocations, and GPA also confirmed that elastic strain is still present at the interface. The quantification of the strain in the theoretical paper is not precise enough to establish a direct comparison. Nevertheless, the high mismatch of the heterojunction and the observed bending behaviours make this explanation the most likely. Indeed, the reduction of the bandgap when fitting the interface with an indirect pattern is coherent with the reduction of the conduction band minimum

observed in the theoretical paper. Moreover, the band origin of the dislocations acting as low bandgap centres could also be related to the displacement of the conduction band minimum. In addition, fitting a direct pattern does not produce physically possible results, as can be checked in the significance tests at the end of the supporting information.

The following text has been added in the main text to provide insight on this discussion.

In the final paragraph before the conclusions:

“This difference in fitting performance and obtained average band gap values per region suggests a band type transition from direct to indirect of ZnTe under compressive strain (see Figure 5f and further details in Section S18) and supports what was predicted theoretically by DFT calculations in the literature.⁸⁴ From this theoretical study we conclude that the direct interband transition at the Γ point would change from valence Γ to conduction Z (indirect), due to a decrease in the conduction band minimum at this Z point. With its high local accuracy (0.9 nm pixel size) and the tolerance towards Cherenkov radiation, our measurements constitute, to the best of our knowledge, the first evidence in which high sensitivity band gap structure characteristics have been reported experimentally.”

REVIEWERS' COMMENTS

Reviewer #2 (Remarks to the Author):

Dear Editor

Authors have replied exhaustively to questions and have changed parts of the papers according to the most of my and other reviewers' suggestions. Discussion about potential source of errors/possible artefact is extended and raises most of experimental, sample related and EELS data processing aspects of the studies.

This paper presents the results of challenging, first of this kind studies, therefore this paper can be considered as a "reference point" for similar attempts of local bandgap mapping in nanoscale.

I strongly support publication of this paper in Nature Communication in the current form.

Best regards

Sławomir Kret

Reviewer #3 (Remarks to the Author):

All the comments raised by the reviewers in previous rounds have been addressed adequately to my opinion. I think it is high quality paper, and I support publishing it in this journal.